# Does political participation help improve the life satisfaction of urban residents: Empirical evidence from China

**Li He**[1], **Kun Wang** [1], **Tianlan Liu**[1], **Tianyang Li**[1], **Baolin Zhu**[2]*

**1** School of Philosophy, Zhongnan University of Economics and Law, Wuhan, Hubei, China, **2** School of Marxism, Hubei University of Economics, Wuhan, Hubei, China

* baolin12345@163.com

**Data Availability Statement:** The anonymous datasets for this study can be found in the CSS survey data (http://47.97.196.79:8099/skyuser/user_center). Others can access the data by submitting a data request form. The authors did

## Abstract

Studies have shown that political participation does not only affect the flow of public resources but also creates positive feedback on participants' subjective perceptions. However, research on the relationship between political participation and the life satisfaction of Chinese urban residents is relatively scarce. Thus, this study investigates whether political participation helps improve the life satisfaction of Chinese urban residents. Based on 2577 samples of the 2015 Chinese Social Survey this study used the Ordinary least squares model, instrumental variable model, and propensity score matching model to explore the relationship between political participation and the life satisfaction of Chinese urban residents. The empirical results show that political participation can significantly improve the life satisfaction of urban residents. Compared with urban residents without political participation, the life satisfaction of the political participants was 0.145 units higher at a 0.05 level of significance. In addition, this improved effect varied in degree among different groups of urban residents and was more significant for females, members of the Communist Party of China, highly educated, and employed urban residents. In China, there is a significant relationship between political participation and the life satisfaction of urban residents, with the life satisfaction of urban residents improving significantly through political participation. There are differences in the level of this improved effect among different urban residents, and it is more significant for females, highly educated, members of the Communist Party, and employed urban residents. To improve the life satisfaction of Chinese urban residents, it is necessary to further broaden their political participation channels.

## Introduction

Political participation has received increased research attention in recent years. The term political participation refers to "legitimate activities undertaken by ordinary citizens almost exclusively in order to directly influence the selection process of government officials and the policies to be implemented" [1]. Different forms of political participation, such as voting, election-related behaviors, and protest, have been largely discussed in previous studies [2].

not have any special access privileges that others would not have.

**Funding:** LH: National Social Science Foundation of China (Program No. 20CZZ012); The funders had no role in study design, data collection and analysis, decision to publish, or preparation of the manuscript.

**Competing interests:** The authors have declared that no competing interests exist.

However, in addition to the above forms, China has other peculiarities. For example, political participation in China mainly focuses on activities related to government policies and regulations and actively expressing opinions through official channels [3]. Therefore, in this study, reflecting opinions to government departments and petitioning government departments are also included in the forms of political participation.

Advocates of participatory democracy theory argue that engaging in political activities fosters desirable personal and social qualities in democratic citizens [4]. In particular, it can create more opportunities among citizens to learn new skills, expand new knowledge, and establish social relations with others. In this way, political participation develops into an integral part of personal growth and achievement; thus influencing individuals' subjective well-being and their life satisfaction [5]. More importantly, participation in political activities has "procedural utility" independent of the effect of public decision [6]. That is, citizens can derive a sense of satisfaction from participating in and influencing public decisions and the transparency of the political process, even if the end result is not what they expected.

Therefore, does the above point apply to China? To answer this question, we must first recognize that China, a country in period of social transformation, there is a problem of unbalanced urban-rural development. In recent years, although the benefits enjoyed by rural residents, such as in education, culture, and economy, have greatly improved compared with the past, and the relative gap between urban and rural development has gradually decreased because of geographical location, policy support, and other reasons, the gap between urban and rural development still remains high [7]. This gap is reflected in many areas such as the level of economic development, institutional system, infrastructure construction, and cultural and educational levels. This "development gap" may lead to differences in the political participation between urban and rural residents and further affect their daily lives. Specifically, urban residents are often able to participate in more political activities with better economic conditions, stronger participation capabilities, and richer political channels. This can bring them certain objective benefits and psychological satisfaction and further enhance their satisfaction with life. Therefore, taking Chinese urban residents as observation objects to explore the life satisfaction effect of political participation, on the one hand, can carry out a more in-depth and rich discussion; On the other hand, it is also more conducive to testing whether the above views of participatory democracy theory are applicable in China.

At present, research on political participation and the life satisfaction of urban residents is increasing, especially in Europe and America, but the extent such empirical survey exists in developing democratic countries is unknown [8]. Therefore, this study uses Chinese urban residents as the research object and uses specific income, education, and political status—variables to measure economic and social status—to investigate the influence mechanism of political participation on urban residents' life satisfaction. This study has both theoretical and practical contributions:

Theoretically, the research on China is an important supplement and enrichment to the research on the political participation of developing countries. In addition, investigating the influence mechanism of political participation on life satisfaction from material and non-material levels can provide a new perspective to explore the relationship between them. Practically, since the political participation of urban residents can directly reflect the level of grassroots democracy and governance in China, this study can provide support for the development of grassroots democracy and modernization of grassroots governance in China.

## Literature review

Political participation, as the core element of democratic practice, has received an increased research attention. In recent years, scholars from various countries have conducted extensive researches on the relationship between political participation and life satisfaction. However, among these previous studies, researches related to political participation among urban residents are scarce.

Life satisfaction refers to people's evaluation of the overall quality of life [9]. It can be evaluated based on emotions, satisfaction with other people's relationships, achieved goals, self-concept, self-perceived daily life ability, as well as social support and community belonging [10]. Life satisfaction can also reflect experiences that have a positive impact on a person [11]. Judging from the existing literature, there are diverse research opinions concerning political participation and urban residents' life satisfaction.

Most scholars acknowledge that political participation has a positive impact on the life satisfaction of urban residents [12]. Additionally, most studies reveal that material and non-material needs are important for the overall measurement and assessment of people's quality of life [13]: On the one hand, political participation helps to meet the material needs of urban residents. Generally, researchers suggest that financial satisfaction is a sub-construct of well-being [14] and has been identified as a representative predictor of life satisfaction in a free market economy [15, 16]. Urban residents often use political institutions in the city to participate in signing petitions, attending meetings, and contacting public officials. Using this to express personal voices [17], raise dissatisfaction with economic inequality, and improve one's financial satisfaction. On the other hand, political participation can also help meet the non-material needs of urban residents. Urban residents living in an environment with a high standard of living will pursue their own spiritual development through more political participation, meet the existential needs of social integration, and thereby increase life satisfaction [18]. Moreover, political participation has played a positive role in enhancing citizens' trust in the government. There is a direct relationship between government trust and satisfaction. Government trust plays a strong and consistent role in buffering the impact of personal income on life satisfaction, thereby dampening the impact of economic inequality on residents' well-being [19]. In addition, some scholars believe that there is a positive correlation between voluntary participation in political activities and happiness [20–22].

Conversely, other scholars believe that political participation has a negative impact on urban residents' life satisfaction [11]. They believe that interest and participation in political activities may encourage social dissatisfaction, which in turn will weaken the life satisfaction of urban residents [23]. The reasons are as follows: First, unlike the "individualistic state," in a collectivist society, there is a tendency for individuals to make a sacrifice for the benefit of the entire society and refrain from pursuing individual wealth and satisfaction [17]; Second, there is a continuous negative correlation between the cities that are forced to vote and the average level of happiness of their residents This is because forcing voters who do not normally vote to vote will reduce their average happiness [24]. Third, although the traditional form of political participation in elections can help improve the life satisfaction of urban residents, other forms of democratic participation in unconventional activities, such as signing petitions, participating in demonstrations, and public protests, have a significant negative correlation with satisfaction [25].

In addition to the above two different perspectives, there are also some studies that refute the claim that differences in life satisfaction among individuals in specific areas can be attributed to political participation issues [26]. Relevant studies show that differences in life satisfaction do exist, but there is little or no evidence that political participation itself is important

[27]. For example, in the United States and South Korea, individuals' political participation has not significantly led to political or policy changes, therefore, political the produces no significant effect on life satisfaction [17].

From current researches, the reason why urban residents' political participation will have an impact on their life satisfaction is mainly because it enhances people's social capital and social relationships, and promotes people's positive personality, self-efficacy, and sense of control [28]. Additionally, political participation will enhance the social resources and social capital of urban residents, thereby enhancing their welfare and happiness [29]. It can also help to improve the quality of horizontal social organizations, optimize the human relationship network and promote the interaction of people's social relationships; thus becoming an effective way to improve life satisfaction [30]. Some scholars have found that the current urban community is no longer just a living space. It can integrate national and government laws, policies, and various regulations and services that benefit people's livelihood, thereby promoting community residents' participation in governance, rights defense, and awareness-building [31]. In response to this, Liu et al. emphasized the importance of the link between political values and citizens' life satisfaction, demonstrating through empirical research that Chinese citizens with stronger authoritarian ideology and ethnic attachment may report higher levels of life satisfaction. Meanwhile, in authoritarian regimes, citizens' life satisfaction is often the main source of state legitimacy [32]. From the perspective of political participation, it has become an important field for the combination of social empowerment and people's satisfaction effect. That is, the higher the participation of urban residents, the higher their satisfaction [28].

In addition, some controversies surround the causal relationship between urban residents' political participation and life satisfaction. Previous studies have revealed that political participation has a significant impact on the overall level of urban residents' life satisfaction, that is, participating in political elections will increase the overall level of people's satisfaction [33]. However, there is also empirical evidence to support the opposite conclusion. This problem can be explained by the concept of "reverse causality," that is, the higher life satisfaction reported by urban residents who participate in the political process is because urban residents with higher life satisfaction are more likely to participate in political activity. Compared with urban residents with lower satisfaction, residents with higher satisfaction are more inclined to maintain their political status by actively participating in political campaigns or parliaments. Not only are they less critical in politics, on the contrary, they are more concerned about social issues, which promotes political participation [34].

In addition, with the popularity of the Internet and the innovation of social media, online political participation has increasingly become an important form of participation [35–37]. Many scholars have begun to pay attention to the life satisfaction effect of Internet political participation. They argue that technological capabilities have transformed political processes to become more open, level, and clear, thereby weakening old structures and establishing new social and organizational models [38]. This means that the Internet has become a tool for political participation and facilitates all kinds of political participation, from service delivery to decision-making. At the same time, urban residents can engage in low-cost and multi-channel political participation through new media, such as the Internet, and enjoy the "information richness" of political participation [39]. These factors all contribute to their life satisfaction. Other scholars disagree with these postulations. They argue that the Internet simplifies political participation in such a way that many citizens who participate in political activities do not really participate but only remain on the surface. In addition, it also strengthens the centralized control of authority [40] and the spread of phenomena such as "clickism" and "lazyism", which is not conducive to the realization of participants' higher life satisfaction.

## Theoretical basis and research hypothesis

In China, the economic interests of urban residents are closely related to politics or policies, therefore, urban residents have a high demand to ensure their own interests through political participation [41]. At the same time, due to the developed economy and diverse occupations in urban areas, urban residents have more access to political participation opportunities, which is conducive to transforming their political advantages into practical personal benefits. From the perspective of material benefits, urban residents will reflect their preferences and needs into government decisions through political participation, thus promoting the improvement of their life satisfaction [27, 42]. The acquisition of material benefits will further promote urban residents to participate in the governance of public affairs at different levels, thus forming a virtuous cycle. From the perspective of immaterial benefits, due to their higher educational level and cultural attainment, urban residents will not only care about direct economic interests but also pay attention to rights and interests beyond economic gains in the process of political participation, such as participation in public affairs and realization of personal rights and own values. The resulting satisfaction will greatly increase their life satisfaction. Therefore, the following hypothesis is proposed in this paper:

**Hypothesis 1: Political participation helps to improve the life satisfaction of urban residents**

For urban residents with different levels of education, political participation has different effects on their life satisfaction. On the one hand, political participation improves urban residents' life satisfaction by directly giving them a sense of participation and achievement, but the level of improvement varies among urban residents with different education levels. On the other hand, political participation indirectly affects urban residents' life satisfaction by shaping their political efficacy. Political efficacy is the feeling or belief that an individual's political behavior can exert influence on the whole political process. It is regarded as a key link between political participation and more general self-development and self-actualization [43]. Individuals who believe that their political actions can have an impact on the political process, that is, they have a high sense of political efficacy, reported better psychological and social well-being [44], which also means higher levels of life satisfaction. Compared with urban residents with low education level, urban residents with high education levels tend to have a higher sense of political efficacy from political participation [45], thus achieving higher life satisfaction. Therefore, the following hypothesis is proposed in this study:

**Hypothesis 2: Political participation has a more significant improvement effect on the life satisfaction of urban residents with higher education levels than that of urban residents with lower education levels**

Compared with urban residents who are not members of the Communist Party of China (hereinafter referred to as CPC members), urban residents who are CPC members, as a special group among urban residents, not only have a stronger willingness to participate in public affairs but also have better information channels and can contact the government more easily [46]. This means they can engage more deeply in political activities and gain more benefits. First, CPC members feel that they have a greater discourse power in the government and can implement or promote change [47], which greatly improves the self-confidence of urban residents and the satisfaction of serving others. Secondly, the act of political participation itself can also improve the skills of citizens [48]. This skill helps improve citizens' ability to benefit from politics. However, due to the close connection with the political system, CPC members can acquire more civic skills from political participation than non-

CPC members. This brings them more benefits and greater life satisfaction. Finally, as an important status symbol, CPC membership in China is often associated with higher income and better career development [49]. Moreover, political participation is of greater help to the career development of CPC urban residents in the future, thus bringing them more improvement in life satisfaction. Therefore, the following hypothesis is proposed:

**Hypothesis 3: Political participation has a more significant improvement effect on the life satisfaction of urban residents who are CPC members than on that of non-CPC members**

Urban residents without jobs are often accompanied by behavioral or psychological tendencies such as staying away from politics, and lacking confidence in the government. Conversely, urban residents with jobs can improve their personal and political efficacy through political participation. This has a positive effect on improving their life satisfaction. On the one hand, personal efficacy is mainly manifested in self-development and social identity. The workplace is also a place of political learning, and work-based political participation can stimulate the creative value of workers in public decision-making [50]. At the same time, active social participation can make workers at the core of work social network [51], so that they have the opportunity to obtain more resources and realize self-development. In addition, due to their psychological attachment to social groups, workers tend to seek benefits for the work group through political participation, and thus gain a sense of group belonging and social support. On the other hand, urban residents with jobs tend to gain higher political efficacy after political participation. In China, urban residents with jobs not only have high political status but also have more opportunities to participate in the unique political environment [52]. This makes their demands and suggestions more likely to be valued and adopted by government departments. This feeling of being responded to or recognized by the government can help increase their satisfaction with life. Therefore, the following assumptions are made:

**Hypothesis 4: The effect of political participation on the life satisfaction of urban residents with jobs is more significant than that of urban residents without jobs**

## Materials and methods

### Data

The data used in this study were drawn from the 2015 Chinese Social Survey (CSS), which was conducted by the Chinese Academy of Social Sciences. The survey adopted multi-stage stratified sampling, and the survey area covered 31 provinces. A total of 10,243 respondents were selected, which was well representative of the total population. According to the applicability of the research question, this study restricted the research participants to Chinese citizens living in urban areas. During the data cleaning process, the number of urban residents who fit the research theme was 2,703, of which a total of 126 respondents accounting for 4.7% were excluded due to missing values. After excluding respondents with missing values, the effective sample size was 2577. To test the effect of missing values, we drew a line graph of the dependent variable before and after removing missing values (Fig 1), and found that the frequency distribution of life satisfaction after removing missing values basically coincided with the frequency distribution before removing missing values. This suggests that missing values in the data are random and have a limited impact on the external validity of the empirical findings.

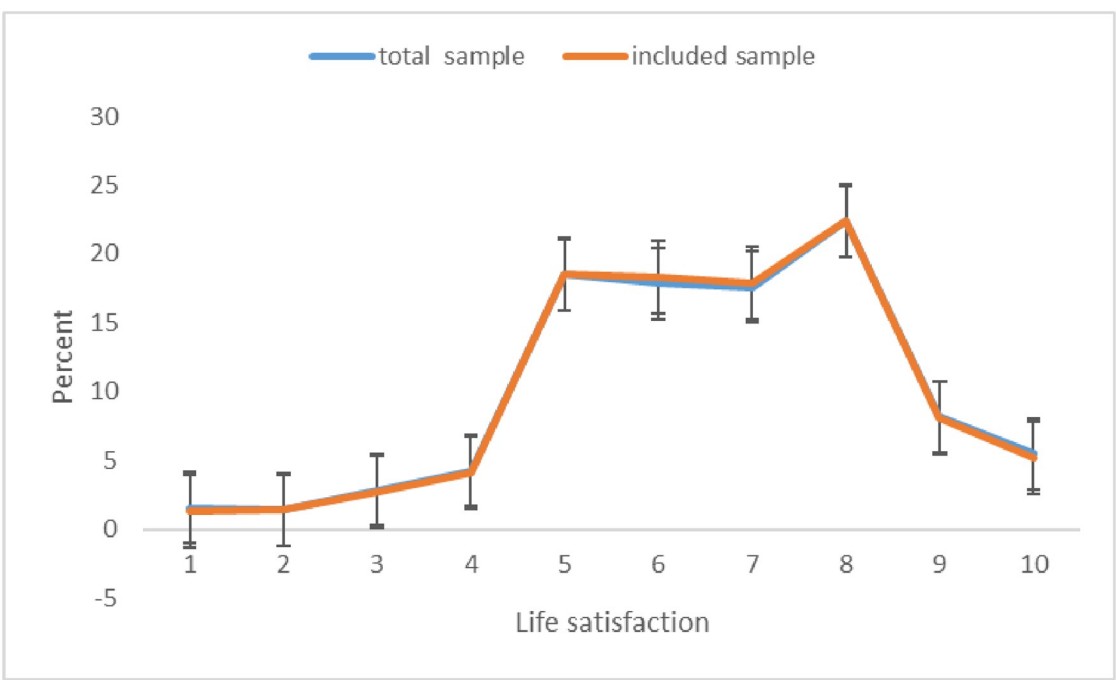

**Fig 1. Comparison of the life satisfaction distribution.**

### Measure

**Dependent variable.** The dependent variable in this study was life satisfaction. Referring to the study of Yang et al [53], this variable was obtained from the CSS question "In general, how satisfied are you with your life." In terms of the variable assignment, we assigned a value from 1 to 10 to life satisfaction based on the respondents' responses. The higher the score, the better the life satisfaction status.

**Independent variable.** The independent variable in this study was political participation. CSS paid great attention to the political participation status of interviewees. The political participation issues include the following seven items: "discuss political issues with others," "reflect social issues to newspapers, radio, and other media," "reflect opinions to government departments," "participate in volunteer activities organized by the government/Danwei/ school," "participate in village/neighborhood committee elections," "petition to government departments," and "participate in demonstrations, strikes, etc." In terms of specific assignments, this study treated individuals who had never participated in any of the above activities as non-political participants and were assigned a value of 0; individuals who had participated in one or more activities were political participants and were assigned a value of 1.

**Control variables.** Focusing on the factors that affect the life satisfaction and referring to the study of Bialowolski and Weziak-Bialowolska [54], for the selection of control variables, this study chose gender, age, marital status, education, and ethnicity as demographic variables; religious belief, CPC, employment, sense of fairness, income, and social trust as social attribute variables, and the province as a regional variable.

### Model selection

**Ordinary least squares (OLS) model.** To explore the effect of political participation on urban residents' life satisfaction, this study first used an OLS model for preliminary estimation.

The model was set up as follows.

$$satisfaction_i = \alpha_0 + \alpha_1 \ politics_i + \alpha_2 \ X_i + \varepsilon_i \tag{1}$$

Where $satisfaction_i$ indicates the life satisfaction of the ith respondent; $politics_i$ indicates whether the ith respondent has ever participated in political activities. $X_i$ represents other control variables; $\varepsilon_i$ represents the random error term; $\alpha_1$ is the coefficient to be estimated in this study, which reflects the magnitude and direction of the influence of political participation on urban residents' life satisfaction.

**Instrumental variable model.** To solve the possible reverse causality problem, this study further adopted the instrumental variable method. There may be mutual influences between urban residents' political participation and life satisfaction. Political participation will affect the life satisfaction of urban residents, and at the same time, individuals with different life satisfaction levels will have different willingness to participate in political participation. For reverse causality problems, it is common practice to find effective instrumental variables. Given that the instrumental variables need to satisfy the requirement that they are related to the independent variable and not directly related to the dependent variable, this study selects the perception of corruption level and perception of national achievement as instrumental variables for the following reasons: On the one hand, the perception of corruption level and perception of national achievement can have a direct impact on the willingness of urban residents to participate in politics, and there is a strong correlation with the independent variable— political participation; On the other hand, since the level of corruption and perception of national achievements are both relatively objective evaluations of macro issues by individuals, they are closely related to the microscopic level of life satisfaction. The direct correlation degree of individual feeling is weak, which meets the requirement that the instrumental variable does not directly affect the dependent variable. In terms of variable measurement, the level of corruption perception is obtained by respondents' responses to "Do you think corruption is the main problem in today's society", and the value of the answer that believes that corruption is the main problem is 1, otherwise it is 0; the perception of national achievement with the statement "I am proud of my country's achievements" was measured, and the responses were divided into "strongly disagree", "somewhat disagree", "somewhat agree" and "strongly agree". Low to high is assigned as 1, 2, 3, 4. The instrumental variable model is set as follows:

$$politics_i = \beta_0 + \beta_1 z_i + \beta_2 X_i + \varepsilon_i \tag{2}$$

$$satisfaction_i = \alpha_0 + \alpha_1 \widehat{politics}_i + \alpha_2 X_i + \varepsilon_i \tag{3}$$

Among them, $z_i$ in Eq (2) is the instrumental variable—perception of corruption and national achievement, and $\widehat{politics}_i$ in Eq (3) is the estimated value of the endogenous variable political participation obtained by Eq (2).

**PSM model.** The political participation behavior of urban residents is not random but will be affected by a variety of factors. This means that political participation may have a self-selection problem, and the difference in life satisfaction between political and non-political participants may come from factors that affect urban residents' political participation rather than political participation itself. Therefore, if we only use OLS regression, it is difficult to obtain unbiased estimation results. In this regard, this study further used the propensity score matching (PSM) method proposed by Rosenbaum and Rubin [55] to test the relationship between urban residents' political participation and their life satisfaction. The basic principle of PSM was to use the individual's propensity to receive treatment to re-screen samples, so as to eliminate the selective deviation between the treatment group and the control group as

much as possible. Specifically, in this study, a logit model was chosen to estimate the propensity score of individuals, and the matching methods chosen were nearest neighbor matching, kernel matching, and radius matching. The specific operation steps are as follows:

$$PS(\boldsymbol{X}) = P_r\{D = 1|\boldsymbol{X}\} = E\{D|\boldsymbol{X}\} \tag{4}$$

Among these, D is the dummy variable of whether urban residents participate in politics. If the respondent has political participation behavior, then D = 1, otherwise, D = 0. X is a covariate that affects individuals' political participation. Given X, the probability of political participation by city residents is equal.

In the second step, three methods of nearest neighbor matching, kernel matching, and radius matching were used to match the treatment group and the control group according to the propensity score.

The third step is to calculate the average treatment effect ATT of the treatment group.

$$ATT = E(satisfaction_1|D = 1) - E(satisfaction_0|D = 0) \tag{5}$$

In this, $satisfaction_1$ represents the life satisfaction score of the individual who is a political participant, $satisfaction_0$ represents the life satisfaction score of the non-political participant, ATT is the life satisfaction score of the political participant and the life satisfaction score assuming that the individual is not a political participant. The difference is the net effect of political participation on the life satisfaction of urban residents.

## Results

### Descriptive statistics

Table 1 reports the descriptive statistical results of the sample. The average life satisfaction score of the sample was 6.6, which was in a relatively happy state. Urban residents who had ever participated in political activities accounted for 55.6% of the total, indicating that there was still a lot of room for expansion of Chinese urban residents' political participation. In terms of other characteristics of the sample, 48.2% were male; the average age was 45.8 years; 80.3% were married; 18.8% were members of the Communist Party of China.

**Table 1. Variable definition and descriptive statistics.**

| Variable | Variable definition | Obs | Mean | Std. Dev. | Min | Max |
|---|---|---|---|---|---|---|
| Life satisfaction | Assign a value from 1–10; the higher the score, the more satisfied with life | 2577 | 6.607 | 1.845 | 1 | 10 |
| Political participation | Yes = 1, No = 0 | 2577 | 0.556 | 0.497 | 0 | 1 |
| Gender | Male = 1, Female = 0 | 2577 | 0.482 | 0.5 | 0 | 1 |
| Age | Actual Age of Respondents | 2577 | 45.838 | 13.561 | 18 | 70 |
| Marital status | Married = 1, Unmarried = 0 | 2577 | 0.803 | 0.398 | 0 | 1 |
| Education | Respondents' years of education | 2577 | 11.39 | 4.08 | 0 | 19 |
| Religious beliefs | Yes = 1, No = 0 | 2577 | 0.125 | 0.33 | 0 | 1 |
| Ethnicity | Han ethnicity = 1, Minority ethnicity = 0 | 2577 | 0.942 | 0.233 | 0 | 1 |
| CPC | Member of the Communist Party of China = 1, No = 0 | 2577 | 0.188 | 0.391 | 0 | 1 |
| Employment | Yes = 1, No = 0 | 2577 | 0.634 | 0.482 | 0 | 1 |
| Income | Respondent's personal total income last year, transformed by the logarithm | 2577 | 10.221 | 1.05 | 4.382 | 14.914 |
| Sense of fairness | Perception of social fairness, the higher the score, the fairer the perception of social fairness | 2577 | 2.665 | 0.557 | 1 | 4 |
| **Social trust** | Perception of social trust, the higher the score, the fairer the society is considered | 2577 | 5.337 | 1.677 | 1 | 10 |
| Province | Eastern province = 1, Other province = 0 | 2577 | 0.404 | 0.491 | 0 | 1 |

## Benchmark regression

Table 2 shows the benchmark regression results. In the Table, Model 1 is the regression result of the life satisfaction to political participation; Model 2 controls the demographic variables on the basis of Model 1; Model 3 controls the social attribute variables on the basis of Model 2; Model 4 adds a regional variable to model 3. The interpretation of the results is based on Model 4. It could be seen that compared with urban residents without political participation, urban residents with political participation were 0.145 units higher in life satisfaction at a 5% level of significance. This is preliminary proof that hypothesis 1 holds.

The estimation results of the control variables are analyzed in this section. In terms of demographic characteristics, life satisfaction was significantly higher in married urban residents than in unmarried urban residents; with increasing education level, the life satisfaction of urban residents also increased accordingly; while gender, age, and ethnicity were not significantly associated with life satisfaction of urban residents. In terms of social attribute characteristics, CPC membership had a significant positive impact on urban residents' life satisfaction; unemployment significantly reduced the life satisfaction of urban residents; increased incomes

**Table 2. Benchmark regression results.**

|  | Model 1 | Model 2 | Model 3 | Model 4 |
|---|---|---|---|---|
| Politics | 0.272*** | 0.126* | 0.131* | 0.128* |
|  | (0.074) | (0.074) | (0.072) | (0.072) |
| Gender |  | -0.010 | -0.050 | -0.044 |
|  |  | (0.072) | (0.073) | (0.073) |
| Age |  | 0.009*** | -0.001 | -0.001 |
|  |  | (0.003) | (0.003) | (0.003) |
| Marital status |  | 0.248** | 0.284*** | 0.279*** |
|  |  | (0.101) | (0.101) | (0.101) |
| Education |  | 0.097*** | 0.074*** | 0.073*** |
|  |  | (0.010) | (0.011) | (0.011) |
| Ethnicity |  | -0.167 | -0.109 | -0.141 |
|  |  | (0.160) | (0.154) | (0.155) |
| Religious beliefs |  |  | 0.025 | 0.013 |
|  |  |  | (0.108) | (0.108) |
| CPC |  |  | 0.324*** | 0.319*** |
|  |  |  | (0.094) | (0.094) |
| Employment |  |  | -0.288*** | -0.278*** |
|  |  |  | (0.091) | (0.091) |
| Sense of fairness |  |  | 0.680*** | 0.680*** |
|  |  |  | (0.071) | (0.071) |
| Income |  |  | 0.164*** | 0.155*** |
|  |  |  | (0.043) | (0.043) |
| Province |  |  |  | 0.138* |
|  |  |  |  | (0.071) |
| _cons | 6.455*** | 4.963*** | 2.277*** | 2.362*** |
|  | (0.057) | (0.265) | (0.500) | (0.500) |
| N | 2577 | 2577 | 2577 | 2577 |
| r2 | 0.005 | 0.046 | 0.102 | 0.103 |

Note: Robust standard errors are in parentheses;

*, **, *** indicate significance at the 10%, 5%, and 1% levels, respectively.

**Table 3. Two-stage least squares regression results.**

|  | Model 5 | Model 6 |
|---|---|---|
|  | Politics | Satisfaction |
| Politics |  | 1.788** |
|  |  | (0.841) |
| Perceived corruption level | 0.043* |  |
|  | (0.023) |  |
| Perception of state achievement | 0.060*** |  |
|  | (0.015) |  |
| Control variables | yes | yes |
| F statistic | 10.29 |  |
| Sargan p | 0.244 |  |
| N | 2515 | 2515 |

and a sense of fairness led to greater satisfaction with life for urban residents. Urban residents who have higher levels of social trust have higher levels of life satisfaction. In addition, urban residents in the eastern provinces had more positive evaluations of their lives compared to those in the non-eastern provinces.

## Endogenous test

Table 3 reports the results of the two-stage least squares regression using instrumental variables. Model 5 is the first stage regression result. The results show that the regression coefficients of the perception of corruption and national achievement significantly reject the null hypothesis, proving that there is a strong correlation between instrumental variables and endogenous explanatory variables. The F-statistic value for identifying weak instrumental variables is greater than 10, rejecting the hypothesis of weak instrumental variables. However, the Sargan statistic p value in the over-identification test is greater than 0.1, which confirms the null hypothesis that all exogenous variables are not correlated with the random error term in the equation, indicating the exogenousness of instrumental variables. Model 6 is the regression result of the second stage. In this result, the estimated value of political participation is positively correlated with life satisfaction and is significant at the 1% level, proving that political participation can significantly improve the life satisfaction of urban residents. This result is consistent with the findings in the benchmark regression, indicating that political participation has a significant positive effect on the life satisfaction of urban residents.

## Robustness test

The link between political participation and the life satisfaction of urban residents might be the result of differences between political and non-political participants in numerous ways. Therefore, there was a selectivity bias. To solve this problem, we further used PSM to examine the relationship between political participation and life satisfaction. Urban residents with political participation were treated as the treatment group, and those without political participation were treated as the control group. Satisfying the balance test is a necessary prerequisite for using PSM, therefore, we need to perform the balance test first. Table 4 shows the results of the balance test, from which it could be seen that most of the covariates had a deviation of less than 10% after matching, proving that the balance test was passed. After that, three matching methods were used in this study: nearest neighbor matching, kernel matching, and radius matching, respectively.

**Table 4. Balance test.**

| Variable | Unmatched | Mean | Bias% | Reduced | t-value | p-value |
|---|---|---|---|---|---|---|
| | Matched | Treated Control | | Bias% | | |
| Gender | U | 0.528 0.424 | 21.0 | | 5.29 | 0.000 |
| | M | 0.528 0.539 | -2.1 | 90.0 | -0.56 | 0.574 |
| Age | U | 45.595 46.142 | -4.0 | | -1.02 | 0.309 |
| | M | 45.607 46.020 | -3.0 | 24.5 | -0.81 | 0.418 |
| Marital status | U | 0.805 0.802 | 0.8 | | 0.19 | 0.848 |
| | M | 0.804 0.815 | -2.6 | -245.7 | -0.71 | 0.475 |
| Education | U | 12.088 10.515 | 39.2 | | 9.91 | 0.000 |
| | M | 12.092 12.010 | 2.0 | 94.8 | 0.56 | 0.573 |
| Religious beliefs | U | 0.121 0.129 | -2.1 | | -0.54 | 0.589 |
| | M | 0.120 0.153 | -9.9 | -364.4 | -2.56 | 0.011 |
| Ethnicity | U | 0.943 0.941 | 1.2 | | 0.31 | 0.753 |
| | M | 0.944 0.921 | 9.9 | -690.9 | 2.46 | 0.014 |
| CPC | U | 0.246 0.115 | 34.5 | | 8.57 | 0.000 |
| | M | 0.246 0.236 | 2.8 | 92.0 | 0.66 | 0.512 |
| Employment | U | 0.654 0.610 | 9.1 | | 2.29 | 0.022 |
| | M | 0.653 0.642 | 2.3 | 74.4 | 0.63 | 0.531 |
| Income | U | 10.262 10.169 | 8.9 | | 2.23 | 0.026 |
| | M | 10.261 10.203 | 5.5 | 37.6 | 1.43 | 0.153 |
| Sense of fairness | U | 2.655 2.678 | -4.3 | | -1.07 | 0.282 |
| | M | 2.655 2.697 | -7.5 | -76.5 | -2.04 | 0.041 |
| Province | U | 0.419 0.385 | 6.8 | | 1.71 | 0.088 |
| | M | 0.419 0.447 | -5.6 | 17.9 | -1.47 | 0.141 |

Table 5 reports the PSM results. The nearest neighbor matching results showed that the life satisfaction score of the treatment group was 0.205 units higher than that of the control group, and it was significant at the 10% level. Comparing the results of kernel matching and radius matching, the three results were highly similar and consistent with the original conclusion. Thus, political participation still had a significant impact on urban residents' life satisfaction after the self-selection problem had been solved.

## Heterogeneity analysis

The life satisfaction effect of political participation may vary among urban residents. Drawing on Pirralha [56], this study further analyzed the impact of political participation on urban residents' life satisfaction under different factors such as education and political identity.

**Table 5. Mean treatment effect of treatment groups.**

| Matching method | Treatment group | Control group | ATT | Bootstrap standard error | t-value |
|---|---|---|---|---|---|
| Nearest neighbor match | 6.727 | 6.521 | 0.205* | 0.111 | 2.00 |
| Kernel match | 6.727 | 6.570 | 0.157* | 0.074 | 1.98 |
| Radius match | 6.725 | 6.522 | 0.204* | 0.111 | 1.98 |

Note:

* indicates significance at the 10% levels; the standard error after matching in row 5 was obtained by bootstrap calculation 500 times.

**Table 6. Heterogeneity analysis.**

| | By education | | By CPC | | By employment | |
|---|---|---|---|---|---|---|
| | Low education | High education | Yes | No | Yes | No |
| Politics | 0.067 | 0.245** | 0.326* | 0.101 | 0.201** | -0.001 |
| | (0.089) | (0.117) | (0.175) | (0.080) | (0.089) | (0.123) |
| Control variables | yes | yes | yes | yes | yes | yes |
| N | 1774 | 803 | 485 | 2092 | 1635 | 942 |
| r2 | 0.090 | 0.100 | 0.134 | 0.076 | 0.124 | 0.090 |

*Note. Robust standard errors are in parentheses; *, **, *** indicate significance at the 10%, 5%, and 1% levels, respectively.

In terms of gender, political participation by female urban residents had a significant impact on their life satisfaction, while this was not the case for male urban residents. As for political identity, the life satisfaction effect of political participation was mainly found among CPC members. The level of education also had an impact on the relationship between political participation and life satisfaction. Political participation by urban residents with low educational attainment had no significant effect on their life satisfaction, while urban residents with high educational attainment increased their life satisfaction through political participation. Moreover, political participation by employed urban residents had a positive effect on their life satisfaction, but this effect was not present among unemployed urban residents (see Table 6).

## Further analysis

Because of the urban-rural dual structure in China and the huge development gap between urban and rural areas, urban residents usually have more political discourse and channels for political participation than rural residents. In light of this, would the effect of political participation be more significant for urban residents than for rural residents? To answer this question, we temporarily included the non-urban population in the sample. Then, we compared the effect of political participation of urban and rural residents and found that the impact of political participation on life satisfaction was more significant for urban residents in comparison to rural residents (see Table 7).

Furthermore, we examine this question, considering that the intensity of political participation may have different effects on life satisfaction. We believe that the scope of political participation is an effective way to measure the strength of an individual's political participation. According to the number of political participation items of individuals, individuals with 1–2 items are mild political participants, while those with 3–7 items are heavy and extensive

**Table 7. Comparison between urban and rural residents.**

| | (1) | (2) |
|---|---|---|
| | Urban | Rural |
| Politics | 0.128* | 0.114** |
| | (0.072) | (0.055) |
| Control variables | Yes | Yes |
| N | 2577 | 5606 |
| r2 | 0.103 | 0.081 |

*Note. Robust standard errors are in parentheses; * and ** indicate significance at the 10% and 5% levels, respectively.

**Table 8. Varying degrees of political participation.**

| | Mild political participation | Heavy political participation |
|---|---|---|
| | Satisfaction | Satisfaction |
| Politics | 0.158** | 0.136 |
| | (0.071) | (0.124) |
| Control variables | yes | Yes |
| N | 2366 | 1355 |
| r2 | 0.180 | 0.177 |

*Note. Robust standard errors are in parentheses;

** indicates significance at the 5% level.

political participants, compared with non-political participants. Table 8 reports the results of different ranges of political participation on life satisfaction. In each model, political actors of the corresponding type are assigned a value of 1, and non-political actors are assigned a value of 0. It can be seen that mild political participation has a significant positive effect on life satisfaction, while heavy political participation has no significant effect on life satisfaction.

## Discussion

### Political participation helps to improve the life satisfaction of urban residents

Chinese society is a typical society with a dual structure of urban and rural areas, and the division of urban and rural areas means the imbalance in resource distribution. China's urban areas enjoy more democratic resources, and urban residents have more opportunities and possibilities to participate in direct democratic activities. Direct democracy can improve the degree of positivity and satisfaction in people's lives, which often leads to a better life [57]. In other words, people living in more democratic areas have higher life satisfaction [58–60]. Therefore, it can be inferred that urban residents living in Chinese urban areas with a high degree of democracy gain benefit through political participation and other forms of direct democracy, thus improving their life satisfaction.

To be specific, citizens' political participation can more or less have an impact on government decision-making and attract certain material benefits. Chinese citizens' political participation is a cost-benefit balancing behavior of rational decision-makers [61], and citizens often carry out political participation activities with certain goals. For urban residents living in areas with higher economic development, after their basic needs are satisfied, they will put forward higher requirements for the quality of life, and they are good at realizing these needs and preferences through political participation channels, especially promoting the formulation and implementation of relevant policies, such as encouraging more local government spending [62], and increasing public investment in infrastructure [63]. In fact, policies that tend to favor the interests of the majority of citizens are indeed the kind of proposals that can have a significant impact on the quality of citizens' daily lives [64]. Political participation promotes the connection between urban residents and the government's ruling process and allows urban residents to directly or indirectly convey issues of interest and concern to the government. The acquisition of these material benefits can greatly improve the life satisfaction of urban residents and become their follow-up motivation for political participation.

In addition, urban residents can get more immaterial benefits through political participation. Political participation may be influenced by individual social and psychological

conditions, including values, sense of efficacy, and participation identification. Moreover, citizens of different social and economic status will also have different political participation activities due to the different resources they occupy [65]. Due to their higher income level and political and cultural attainment, urban residents pay more attention to rights and interests beyond economic interests, such as community safety, quality living environment, and citizen's right to know, in the process of political participation. In other words, urban residents tend to pay more attention to public affairs beyond themselves and are willing to make personal efforts to seek public interests, which is manifested by caring about public policies, participating in political activities, and demanding government accountability [66]. At the same time, participating in public welfare activities may also have the significance of "status symbol" for them, which also urges them to participate in civic activities to satisfy their sense of status. Although the political participation of urban residents is also uncertain, and they do not want to lose their own interests due to participation, it is highly possible for them to obtain immaterial benefits and improve their life satisfaction.

## The improvement effect of political participation on the life satisfaction of urban residents with higher education level is more significant than that of urban residents with lower education level

In general, the influence of political participation on urban residents' life satisfaction has two aspects: direct and indirect aspects.

In terms of the direct aspect, urban residents with higher education level can obtain a greater sense of participation, happiness, and life satisfaction due to their stronger ability to participate in decision-making activities. Psychological and sociological research suggests that perceived control and the ability to participate in decision-making activities are strong predictors of well-being and life satisfaction. Participatory decision-making contributes to people's mental health and job satisfaction [67–69]. Therefore, political participation improves the life satisfaction of urban residents by expanding the channels of participatory decision-making. For urban residents with higher levels of education, especially, their ability and willingness to participate in decision-making activities tend to be stronger, and their political participation has a greater impact. A greater sense of engagement and fulfillment leads them to enjoy higher life satisfaction than less educated urban residents.

Indirectly, urban residents with higher education levels can acquire a stronger sense of political efficacy, which helps to improve their life satisfaction. It is generally believed that political efficacy has a direct influence on political participation. However, studies have shown that voting and participation in election campaigns also have a positive impact on political efficacy [70]. That is to say, political participation may also improve participants' political efficacy. Political efficacy is not produced in a vacuum, but a subjective self-judgment of the interaction between oneself and the political environment based on certain experiences or information. This subjective self-judgment means that political efficacy can have a certain impact on individual subjective feelings including life satisfaction. However, the degree of influence varies greatly among different groups. Lassen and Serritzlew [71] held that when the political system becomes larger, the gap in political effectiveness between highly educated and poorly educated citizens is likely to widen. In urban areas with more complete political systems, this characteristic also exists naturally. Because highly educated urban residents have the ability to interpret and understand politics, they tend to have a higher sense of political efficacy compared with less educated urban residents [45]. On the one hand, political efficacy affects individuals' perception of their own value. The stronger the sense of political efficacy is, the more effective the individual thinks his political participation is [72]. This is an affirmation of their own value

and can improve their life satisfaction. On the other hand, political efficacy enables individuals to achieve self-actualization. Scholars of participatory democracy deem that political efficacy is the internal mechanism of political participation and self-realization. Individuals achieve self-actualization through having a voice in the government or social/political participation in actions such as freedom of speech [56], thus achieving a higher level of personal life satisfaction [73]. In conclusion, although political participation can generally improve the life satisfaction of urban residents, urban residents with higher education level have stronger political participation ability and political efficacy, and their degree of improvement is more significant than urban residents with lower education level.

## The improvement effect of political participation on the life satisfaction of urban residents who are CPC members is more significant than on that of urban residents who are not CPC members

CPC membership is often accompanied by higher levels of political knowledge [3], which means greater access to political information and the ability to engage in more and deeper political activities. Therefore, for urban residents who are CPC members, political participation has a higher frequency, deeper degree, and greater benefits, and the improvement effect of political participation on their life satisfaction is more significant than that of non-CPC members.

First, CPC membership often leads to a higher sense of political efficacy [3], which makes urban residents more likely to participate in politics and realize their political rights. People contribute their resources and potential to the political process and obtain psychological satisfaction through participation activities [74]. To be specific, on the one hand, CPC members in the urban area contribute in the process of political participation by promoting decision implementation to convince others to their own ability; On the other hand, when individuals choose to engage in certain behaviors by expressing their political values, such behaviors can give them a sense of autonomy that contributes to their overall subjective happiness [75]. For urban CPC members, this sense of autonomy to serve others and promote development can greatly improve their happiness and life satisfaction.

Second, political participation improves the professional civic skills of CPC members in urban areas. Participation in voting and advocacy may encourage the development of better civic skills [76]. Additionally, most of these civic skills are professional skills that promote citizens' better participation in public political affairs. For urban residents who are CPC members, on the one hand, due to their deeper political participation, the improvement effect of acquiring civic skills will be greater; improved civic skills help individuals gain a greater sense of empowerment and self-determination through their interaction with the government system, which leads to higher levels of life satisfaction. On the other hand, such professional skills, which can enhance citizens' participation in public political affairs, are more useful to CPC members in urban areas. It will not only promote them to engage in higher-quality political participation activities but also provide greater help for their future development. However, urban residents who are not CPC members have less demand for such professional civic skills. Therefore, political participation has a more significant effect on CPC members' life satisfaction in urban areas.

Finally, political participation provides CPC members with greater career development opportunities. In China, CPC membership is almost a prerequisite for work and promotion in government agencies [49]. In addition, CPC members have more access to senior officials and are more familiar with government policies. In such an environment conducive to participation in decision-making, personal career aspirations will be more easily realized [5]. In other

words, in the process of political participation, urban residents with CPC membership will not only obtain their own development but will also continuously participate in government decision-making, accumulate political experience, and expand social connections, so as to provide help for their future career development. In conclusion, political participation, as a way to achieve career development, can bring greater development space and opportunities to CPC members in urban areas, which will significantly improve their life satisfaction.

## The effect of political participation on the life satisfaction of urban residents with jobs is more significant than that of urban residents without jobs

Unemployed urban residents are less politically engaged than working urbanites. Unemployment leads to the chronic exclusion of urban residents from the labor market and hinders their social integration [21, 77]. Unemployed urban residents lack resources and political identity, and at the same time, distrust the government's inability to address unemployment-related problems [78–81]. As such, they tend to maintain a state of apathy and alienation from mainstream politics [82, 83]. On the contrary, urban residents who have jobs often show a strong interest in politics and are willing to actively participate in political activities, thereby further improving their life satisfaction. This can be explained from the perspective of personal efficacy and political efficacy.

On the one hand, urban residents with jobs participate in political activities in their job roles, which in turn enhances their personal efficacy, that is, self-development and social identity. In terms of self-development, the participation of working urban residents in political activities do not only cultivate their ideal qualities of democratic citizens [4], but also enable them to obtain resources and opportunities for personal growth and self-actualization, and thus have a positive effect on their personal lives [56]. Political participation and attitudes in the workplace stem in part from the selection process, and participation and attitudes become part of the measures that determine whether they will eventually be able to occupy positions of influence. Individuals who tend to be politically involved are more likely to network with influential colleagues in the workplace through social interactions. This is positively correlated with life satisfaction. In terms of social identity, the higher political participation of urban residents with jobs provides them with a common identity and a sense of community belonging. In China, the employment unit is one of the main places where urban residents form social groups [84]. Actively participating in work-related political activities is motivated by the sense of responsibility conferred by the role of the work and the pursuit of the common interests of the group. Precisely, political participation can further enable workers to gain the identification of the workgroup and other social support. This helps to increase their life satisfaction.

On the other hand, workers' political status and workplace participation mechanisms provide opportunities for developing the political efficacy of working urban residents [85], which in turn has a positive effect on their life satisfaction. As mentioned above, political efficacy is a person's ability to believe that his or her political participation affects the political system and government decision-making. China is a socialist country. The constitution and laws endow workers with lofty political status and provide them with abundant channels for political participation, such as people's congresses at all levels and trade unions. In addition, a worker is able to extend his experience in his job to the political arena through democratic learning on the job, including familiarity with democratic procedures and political democratic skills [86]. Precisely, workers' higher political status and mature political participation skills make it easier for their opinions and suggestions to be valued or even adopted by authorities. This increases their sense of political efficacy as well as their life satisfaction.

## Conclusion

Using CSS2015 data, this study analyzed the impact of political participation on the life satisfaction of Chinese urban residents. The findings indicate that political participation increases the life satisfaction of Chinese urban residents. Specifically, the life satisfaction was 0.145 units higher at a 5% level of significance for urban residents with political participation compared to those without political participation. It is noteworthy that the magnitude of this impact varied across different groups of urban residents. The more affected groups of urban residents were characterized as highly educated, Chinese Communist Party members, and employed. In addition, the life satisfaction effect of political participation was more significant for urban residents compared to rural residents.

Based on the above findings, to improve the life satisfaction of urban residents, we suggest the following. First, the government should pay more attention to increasing the political participation rate of urban residents. This can be done in two ways: promoting urban grassroots autonomy and broadening the channels of citizens' political participation. On the one hand, grassroots self-governance can enhance urban citizens' intrinsic sense of efficacy, train their ability to handle public affairs, and further increase urban residents' subjective willingness to engage in political participation. On the other hand, smooth political participation channels can raise urban residents' expectations of the rewards of political participation and lower their costs of political participation, which in turn drives rational urban residents to engage in political activities. In terms of specific policy directions, the government should return the power of community governance to urban residents, as well as loosen its control over social organizations. Second, urban residents should also increase their interest in political participation. The effective operation of a democratic system cannot be achieved without the active political participation of citizens. Further, the lack of citizens' awareness of political participation and ability to handle social affairs may affect the proper functioning of the democratic system. A poorly functioning system will in turn alienate citizens from politics, thus creating a vicious circle. Therefore, urban residents should firmly establish a sense of ownership and realize that political participation is not only their right but also their duty. In the meantime, urban residents should consciously develop their capacity for consultation and negotiation to participate more effectively in the governance process of public affairs.

The limitations of this study exist in the following three aspects. First, the data used are cross-sectional and may be relatively weak in the interpretation of causal mechanisms. Second, the selection of instrumental variables needs to be further improved. Third, the specific mechanisms by which political participation affects urban residents' life satisfaction need to be further explored. These limitations provide directions for further research.

## Acknowledgments

### Ethics approval

This research was approved by Research Ethics Committee, Institute of Sociology, the Chinese Academy of Social Sciences, and all the participants provided signed informed consent at the time of participation. The study methodology was carried out in accordance with approved guidelines.

## Author Contributions

**Conceptualization:** Li He.

**Data curation:** Kun Wang.

**Formal analysis:** Kun Wang.

**Funding acquisition:** Baolin Zhu.

**Investigation:** Baolin Zhu.

**Methodology:** Kun Wang.

**Project administration:** Li He.

**Resources:** Baolin Zhu.

**Software:** Kun Wang.

**Supervision:** Baolin Zhu.

**Validation:** Kun Wang.

**Visualization:** Kun Wang.

**Writing – original draft:** Tianlan Liu, Tianyang Li.

**Writing – review & editing:** Li He, Baolin Zhu.

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
