## [Decision Letter · Decision Letter 0]

19 May 2022

PONE-D-22-02512Does political participation help improve the life satisfaction of urban residents: Empirical evidence from ChinaPLOS ONE

Dear Dr. Zhang,

Thank you for submitting your manuscript to PLOS ONE. After careful consideration, we feel that it has merit but does not fully meet PLOS ONE’s publication criteria as it currently stands. Therefore, we invite you to submit a revised version of the manuscript that addresses the points raised during the review process.

We look forward to receiving your revised manuscript.

Kind regards,

Rong Zhu, Ph.D.

Academic Editor

PLOS ONE

Journal Requirements:

a) Did participants provide their written or verbal informed consent to participate in this study?

b) If consent was verbal, please explain i) why written consent was not obtained, ii) how you documented participant consent, and iii) whether the ethics committees/IRB approved this consent procedure

Reviewers' comments:

Reviewer's Responses to Questions

**Comments to the Author**

1. Is the manuscript technically sound, and do the data support the conclusions?

Reviewer #1: Yes

Reviewer #2: Partly

2. Has the statistical analysis been performed appropriately and rigorously? 

Reviewer #1: Yes

Reviewer #2: N/A

3. Have the authors made all data underlying the findings in their manuscript fully available?

Reviewer #1: Yes

Reviewer #2: Yes

4. Is the manuscript presented in an intelligible fashion and written in standard English?

Reviewer #1: Yes

Reviewer #2: Yes

5. Review Comments to the Author

Reviewer #1: Review on the manuscript

Title: Does political participation help improve the life satisfaction of urban residents: Empirical evidence from China

Journal: PLOS ONE

10.05.2022

The article is dedicated to the discussion of political participation as a factor to improve life satisfaction in urban China. The article is well structured and gives a clear idea of the research question and the appropriate methodology that is used by the authors. The literature review provides a deep analysis of the existing studies and serves as a good ground for the theoretical explanations of the hypotheses.

The article adds to the literature and methodological discussion since it covers the aspect of urban population life satisfaction testing the political participation of educated and non-educated groups of the population and more.

The manuscript can generate wide public interest and attract many readers. It fits the scope of the journal, and it is ready for publication in the journal after addressing a few concerns mentioned below.

Major concerns

•It is not very clear how the authors address the problem of endogeneity they discuss on page 8 (reversed causality). If there is evidence of that in the literature, how do authors address it in their study, and why do they keep the logic of the causality from political participation to life satisfaction?

•Why political participation is measured as a dummy variable? Why not index? Would such an index that counts the number of different political participation (form 7 items) be of better indicator to reflect political participation?

•Probably it will make sense to estimate the logit or probit model since the dependent variable has an ordered scale (not OLS). Especially for the fact to compare it with PSM logit later. Comparison of OLS with PSM logit would not be that correct if compare ordered logit with PSM logit.

•In the Data description section the authors stated that the data has a total of 10243 individuals, and the urban population is 2577, why not have estimations of the non-urban population to see if political participation in rural areas has any impact on life satisfaction? Later on page 20 the authors are placing that comparison actually, but initially, it is not discussed either in the introduction or in the methodology, not even in the data description.

•The authors have analysis for the employment variable but do not have hypotheses on that. Moreover, it is better to organize the results in the same sequence as the hypotheses were discussed. First on education and then CPC…

•What are the possible effects of digitization of society (including online political participation) on life satisfaction could be?

•It is not clear why we need table 5 if the results of the two last columns are never used in discussions or conclusions. Why not just keep Ologit? Moreover, the final results used for the conclusions are from Ologit model, not from PSM – why? it would be nice to have a clear statement in the conclusions about solving the endogeneity problem and use the results of political participation as the predictor of life satisfaction not Ologit, maybe?

•Overall the paper would gain if the text would be tighter in terms of matching between the sections, especially the results and methodology/hypotheses part.

Minor comments

line 213 – better to use word – respondents/individuals – instead of the word – samples

is it possible to control for social capital (trust, friends, relations, social media engagement) since those variables were proved to be strong life satisfaction determinants?

Table 5 – better to name the models accordingly as OLS, Ologit, PSM not 1,2,3 so that the reader can understand better, or give a not which model is what under the table.

Table 6 – why do you need a gender column? The authors did not have a hypothesis on that.

Reviewer #2: This paper is theoretically interesting and practically relevant. However, it needs revisions before in my view publication is possible.

I will list my detailed comments and suggestions below:

The introduction section should be restructured to focus on this paper’s outcome variable.

The author might want to incorporate the most recent studies on life satisfaction in China in their literature review, such as Liu, Xinsheng, Youlang Zhang, and Arnold Vedlitz. "Political Values and Life Satisfaction in China." The China Quarterly 245 (2021): 276-291.

Sentences like “As the core element of democratic 84 operation, political participation has always been a hot topic in academic research at home and abroad.” sound chinglish.

How serious is the missing value problem? Does it affect the external validity of the empirical findings?

The authors may want to clarify the potential bias caused by reverse causality. PSM does not deal with reverse causality.

There are quite a few typos, grammatical errors, and confusing phrases in the manuscript. The authors might want to hire a professional editor to proofread their next version of the manuscript.

6. PLOS authors have the option to publish the peer review history of their article (what does this mean?). If published, this will include your full peer review and any attached files.

Reviewer #1: **Yes: **Tatiana Karabchuk

Reviewer #2: No

---

## [Author Response · Author response to Decision Letter 0]

1 Jul 2022

We deeply appreciate the helpful comments from the two reviewers on the original manuscript. We have carefully considered the comments and have revised the manuscript accordingly. A point-by-point discussion of each comment follows.

Reviewer 1：

Major concerns 

1. It is not very clear how the authors address the problem of endogeneity they discuss on page 8 (reversed causality). If there is evidence of that in the literature, how do authors address it in their study, and why do they keep the logic of the causality from political participation to life satisfaction?

Response: We thank the reviewer for this pertinent observation. We agree with the reviewer that we did not address the issue of endogeneity very well in the original version. Consequently, we re-estimated the relationship between urban residents' political participation and life satisfaction using the instrumental variable method. The basic principle of the instrumental variable method is to estimate the fitted value of the independent variable by using the variable that is related to the independent variable but not related to the instrumental variable and then regressing the dependent variable to this fitted value, thereby solving the problem of reverse causality. According to the requirements of instrumental variables, we selected the perception of corruption degree and national achievement as the instrumental variables for the following reasons: on the one hand, the perception of corruption degree and national achievement perception can have a direct impact on the willingness of urban residents to participate in politics, which is closely related to the independent variable political participation. On the other hand, since the degree of corruption and the perception of national achievement are both relatively objective evaluations of macro issues by individuals, the direct correlation with micro-level individual perceptions such as life satisfaction is weak, and the instrumental variables are not directly related. Therefore, we believe that the perception of corruption level and the perception of national achievement are more appropriate instrumental variables, and the relevant test results also support our judgment. The estimation results of instrumental variables show that after solving the problem of reverse causality, political participation still has a significant effect on the life satisfaction of urban residents. (see page 17).

2. Why political participation is measured as a dummy variable? Why not index? Would such an index that counts the number of different political participation (form 7 items) be of better indicator to reflect political participation?

Response: We thank the reviewer for this valuable observation. The reviewer’s comment has given us new insight that using an index to measure political participation is indeed a viable suggestion. Consequently, in our further analysis, we use the intensity of political participation as a way of measuring the intensity of individual political participation. Moreover, according to the number of respondents' political participation categories, the observation objects are divided into mild political participants (1-2 items) and severe political participants (3-7 items). It is found that mild political participation can significantly improve the life satisfaction of urban residents, while heavy political participation has no significant effect on individual life satisfaction (see page 27, line 480-490). However, considering that our research topic is whether political participation has a significant impact on the life satisfaction of urban residents, and to explore the difference in life satisfaction between urban residents with and without political participation, we still use Dummy variables to measure the political participation of individuals. Of course, in the upcoming study on the impact of political participation, we will more comprehensively adopt the reviewers' opinions and use the number of political participation to measure the degree of political participation.

3. Probably it will make sense to estimate the logit or probit model since the dependent variable has an ordered scale (not OLS). Especially for the fact to compare it with PSM logit later. Comparison of OLS with PSM logit would not be that correct if compare ordered logit with PSM logit.

Response: We thank the reviewer for this pertinent comment. When the dependent variable is an ordinal scale, it makes more sense to use a logit model or a probit model. However, some studies have revealed that when ordinal variables have many assignment intervals, they can also be treated as continuous variables, and the OLS model can be used [1]. Therefore, we use an OLS model that is easier for readers to understand for regression. In addition, the reviewer’s evaluation of the PSM model is biased due to our simplistic introduction to the PSM model. We are sorry for this oversight. In the revised version, we have introduced the PSM model in more detail so that readers can understand the principle and function of the model more clearly.

references:

[1] Ferrer-i-Carbonell A, Frijters P. How Important is Methodology for the Estimates of the Determinants of Happiness? Econ J (2004) 114:641-659. doi: 10.1111/j.1468-0297.2004.00235.x

4. In the Data description section the authors stated that the data has a total of 10243 individuals, and the urban population is 2577, why not have estimations of the non-urban population to see if political participation in rural areas has any impact on life satisfaction? Later on page 20 the authors are placing that comparison actually, but initially, it is not discussed either in the introduction or in the methodology, not even in the data description.

Response: We thank the reviewer for this valuable observation. We apologize for omitting in the introduction a discussion of the differences in political participation between urban and rural residents. In response to the reviewer’s suggestion, we have added a description of the gap between urban and rural areas in China under the dual structure of urban and rural areas in the Introduction section. The gap between urban and rural areas will lead to differences in the way and degree of political participation of urban and rural residents. Compared with rural residents, the more diverse forms and breadth of political participation of urban residents make them more valuable for research and more conducive to testing whether the theory of participatory democracy is applicable in China. Therefore, this study focuses on the impact of political participation on the life satisfaction of urban residents.

5. The authors have analysis for the employment variable but do not have hypotheses on that. Moreover, it is better to organize the results in the same sequence as the hypotheses were discussed. First on education and then CPC…

Response: We thank the reviewer for this valuable suggestion, which will help in filling the gap in our research. We are sorry that we left out assumptions and discussions on employment variables. In response to this, we added Hypothesis 4: The effect of political participation on the life satisfaction of urban residents with jobs is more significant than that of urban residents without jobs, and we systematically discuss this hypothesis at the end. For the order of results, we made adjustments in the Heterogeneity Analysis section. The order of each relevant module is uniformly adjusted to education, CCP membership, and work status.

6. What are the possible effects of digitization of society (including online political participation) on life satisfaction could be?

Response: We thank the reviewer for seeking this clarification, which has greatly broadened our research horizons. The impact of a digital society is a very important topic, and we have a strong research interest in it. To this end, in the introduction part and the literature review part, we report and sort out the Internet political participation at the practical and theoretical levels respectively.

It is a pity that the digitalization of Chinese society has only developed rapidly in recent years, and there is a lack of reliable survey data on Internet political participation. Data constraints prevent us from directly exploring this exciting problem. However, we continue to monitor progress on this issue and will conduct research on Internet political participation when the time comes.

7. It is not clear why we need table 5 if the results of the two last columns are never used in discussions or conclusions. Why not just keep Ologit? Moreover, the final results used for the conclusions are from Ologit model, not from PSM – why? it would be nice to have a clear statement in the conclusions about solving the endogeneity problem and use the results of political participation as the predictor of life satisfaction not Ologit, maybe?

Response: Thanks to the reviewers for their comments. Considering that the content of Table 5 is easy to be misunderstood, and at the same time, we used instrumental variables in the revision to deal with the endogeneity problem, which weakened the importance of its content, we decided to delete Table 5 and related content in it in the main text.

8. Overall the paper would gain if the text would be tighter in terms of matching between the sections, especially the results and methodology/hypotheses part.

Response: We thank the reviewer for this valuable comment. In the original research design and writing, we had many shortcomings, especially, because results and methods were not rigorous enough. The reviewers have suggested many helpful revisions to our paper. We carefully analyzed these comments and made corresponding revisions to the original text to make the structure of the paper more rigorous (see the red part of the article). We believe that with the help of the reviewers, the quality of this paper has been greatly improved. We sincerely thank the reviewers for their valuable insights.

Minor comments 

1. line 213 – better to use word – respondents/individuals – instead of the word – samples

Response: We thank the reviewer for this insightful comment. We have revised this word and carefully checked and revised the wording of the full text, and submitted it to a professional institution for language polishing.

2. is it possible to control for social capital (trust, friends, relations, social media engagement) since those variables were proved to be strong life satisfaction determinants?

Response: We thank the reviewer for this valuable comment. Following the reviewers' comments, and taking into account data availability, we controlled for trust as a social capital variable. The results show that urban residents who have higher levels of social trust have higher levels of life satisfaction. (See page 21, lines 408-409)

3. Table 5 – better to name the models accordingly as OLS, Ologit, PSM not 1,2,3 so that the reader can understand better, or give a not which model is what under the table.

Response: We appreciate the reviewer’s comment here. Considering that the content of Table 5 is easy to be misunderstood, and we used instrumental variables to deal with the endogeneity problem in the modification, thereby weakening the importance of the content of Table 5, we have deleted Table 5 and subsumed its contents into the main text.

4. Table 6 – why do you need a gender column? The authors did not have a hypothesis on that.

Response: We thank the reviewer for this valuable observation. This study was not intended to delve into gender heterogeneity as the reviewer pointed out, therefore, following the reviewer’s comment, we have expunged the gender column in Table 6.

Reviewer 2

1. The introduction section should be restructured to focus on this paper’s outcome variable.

Response: We thank the reviewer for this valuable comment. We have added a discussion on the differences in political participation of urban and rural residents in the Introduction section to highlight the value of research on the relationship between political participation and the life satisfaction of urban residents. Moreover, we have revised the structure of the Introduction in such a way that it corresponds with the research results. 

2. The author might want to incorporate the most recent studies on life satisfaction in China in their literature review, such as Liu, Xinsheng, Youlang Zhang, and Arnold Vedlitz. "Political Values and Life Satisfaction in China." The China Quarterly 245 (2021): 276-291.

Response: We thank the reviewer for this important comment. The reviewer’s comments are very valuable and will enrich our current research. In line with the reviewer’s suggestion, we have added more recent citations on life satisfaction in China by Liu et al. In addition, we have added the following five more recent studies:

[2] Lühr M, Pavlova M K, Luhmann M, “Nonpolitical Versus Political Participation: Longitudinal Associations with Mental Health and Social Well-Being in Different Age Groups,” Social Indicators Research: An International and Interdisciplinary Journal for Quality-of-Life Measurement, Springer, vol. 2022, 159(3): 865-884, February.

[3] Carle J. “Welfare Regimes and Political Activity among Young Unemployed People”, in Hammer T. (ed.), Youth Unemployment and Social Exclusion in Europe. A comparative Study, Bristol: Policy Press, 2003, 193-207.

[4] Dawson-Townsend, K. Social participation patterns and their associations with health and well-being for older adults. SSM - Population Health, 8, 100424.

[5] Hammer T. Mental health and social exclusion among unemployed youth in Scandinavia. A comparative study. International Journal of Social Welfare, 2000, 9(1), 53–63. 

[6] Lawton R N, Gramatki I, Watt W, Fujiwara D. Does volunteering make us happier, or are happier people more likely to volunteer? Addressing the problem of reverse causality when estimating the wellbeing impacts of volunteering. Journal of Happiness Studies, 2020, 22(11), 599–624.

3. Sentences like “As the core element of democratic 84 operation, political participation has always been a hot topic in academic research at home and abroad.” sound chinglish.

Response: We thank the reviewer for this pertinent comment. We have revised the sentence to a better formal tone “Political participation, as the core element of the democratic operation, has received an increased research attention” (see page 5, lines 98-99). Moreover, to improve the English language accuracy and flow of the manuscript the paper has been edited by a professional editing company (Editage). 

4. How serious is the missing value problem? Does it affect the external validity of the empirical findings?

Response: We thank the reviewer for seeking this clarification. During the data exclusion process, the number of urban residents that fit the research theme was 2,703, of which a total of 126 respondents accounting for 4.7% were excluded due to missing values. To test the influence of missing values, we drew a line graph of the dependent variable before and after removing missing values and found that the frequency distribution of life satisfaction after removing missing values basically coincided with the frequency distribution before removing missing values. This suggests that missing values in the data are random and have a limited impact on the external validity of the empirical findings. (see page 14, lines 278-286)

5. The authors may want to clarify the potential bias caused by reverse causality. PSM does not deal with reverse causality.

Response: We thank the reviewer for this pertinent comment. The PSM does not solve the problem of reverse causality. To address this issue, we further re-estimate the relationship between urban residents' political participation and life satisfaction using an instrumental variable approach. The basic principle of the instrumental variable method is to estimate the fitted value of the independent variable by using the variable that is related to the independent variable but not related to the instrumental variable and then regressing the dependent variable to the fitted value, thereby solving the problem of reverse causality. According to the requirements of instrumental variables, we selected two instrumental variables: perception of corruption degree and perception of national achievement. There is a strong correlation between participation; on the other hand, since the degree of corruption and the perception of national achievement are both relatively objective evaluations of individuals on macro issues, they are weakly related to the micro-individual feelings of happiness, and satisfying the instrumental variables does not directly affect the factors. Therefore, we believe that the perception of corruption level and the perception of national achievement are more appropriate instrumental variables, and the relevant test results also support our judgment. The estimation results of instrumental variables show that after solving the problem of reverse causality, political participation still has a significant effect on the life satisfaction of urban residents.

6. There are quite a few typos, grammatical errors, and confusing phrases in the manuscript. The authors might want to hire a professional editor to proofread their next version of the manuscript.

Response: We thank the reviewer for this observation. We have revised the manuscript to address grammatical errors, language issues and awkward construction. Moreover, to improve the English language accuracy of the manuscript, this version has been edited by a professional English editing company (Editage).

---

## [Decision Letter · Decision Letter 1]

2 Aug 2022

PONE-D-22-02512R1Does political participation help improve the life satisfaction of urban residents: Empirical evidence from ChinaPLOS ONE

Dear Dr. Wang,

Thank you for submitting your manuscript to PLOS ONE. One referee is happy with the revised version, while the other request some minor changes. Therefore, we invite you to submit a revised version of the manuscript that addresses the points raised during the review process.

We look forward to receiving your revised manuscript.

Kind regards,

Rong Zhu, Ph.D.

Academic Editor

PLOS ONE

Journal Requirements:

Reviewers' comments:

Reviewer's Responses to Questions

**Comments to the Author**

1. If the authors have adequately addressed your comments raised in a previous round of review and you feel that this manuscript is now acceptable for publication, you may indicate that here to bypass the “Comments to the Author” section, enter your conflict of interest statement in the “Confidential to Editor” section, and submit your "Accept" recommendation.

Reviewer #1: All comments have been addressed

Reviewer #2: (No Response)

2. Is the manuscript technically sound, and do the data support the conclusions?

Reviewer #1: Yes

Reviewer #2: (No Response)

3. Has the statistical analysis been performed appropriately and rigorously? 

Reviewer #1: Yes

Reviewer #2: (No Response)

4. Have the authors made all data underlying the findings in their manuscript fully available?

Reviewer #1: Yes

Reviewer #2: (No Response)

5. Is the manuscript presented in an intelligible fashion and written in standard English?

Reviewer #1: Yes

Reviewer #2: (No Response)

6. Review Comments to the Author

Reviewer #1: The authors did good job by adjusting the text to all the mentioned points. Just one more minor thing is left before publication: proofreading for the grammar mistakes and typo like "Meavy political participation" in Table 8 - means Heavy or involved political participation?

Also PSM - this is the comparison between urban and nonurban? Correct? Why then the data description is made still only for urban sample? As if the non-urban population is cut from the analysis but then it somehow appeared in the results discussion.

The rest is all clear.

Reviewer #2: No further comments.

7. PLOS authors have the option to publish the peer review history of their article (what does this mean?). If published, this will include your full peer review and any attached files.

Reviewer #1: **Yes: **Tatiana Karabchuk

Reviewer #2: No

---

## [Author Response · Author response to Decision Letter 1]

3 Aug 2022

We deeply appreciate the helpful comments from the two reviewers on the original manuscript. We have carefully considered the comments and have revised the manuscript accordingly. A point-by-point discussion of each comment follows.

Reviewer1:

1. The authors did good job by adjusting the text to all the mentioned points. Just one more minor thing is left before publication: proofreading for the grammar mistakes and typo like "Meavy political participation" in Table 8 - means Heavy or involved political participation?

Response: We thank the reviewer for this observation. We have revised the manuscript to address grammatical errors, language issues and awkward construction.

2. Also PSM - this is the comparison between urban and nonurban? Correct? Why then the data description is made still only for urban sample? As if the non-urban population is cut from the analysis but then it somehow appeared in the results discussion.

Response: We thank the reviewer for this pertinent comment. We apologize for the ambiguity in our original manuscript regarding the role of PSM and the comparison between urban and nonurban. PSM compared the difference in life satisfaction between political and non-political participants in urban areas, rather than comparing urban and rural residents. Our empirical analyses were based on an urban sample, with the exception of further analysis concerning urban-rural comparison. The discussion on non-urban population originated from a reviewer's suggestion that we should add this section to the original manuscript. We followed this suggestion and compared the differences in political participation between urban residents and rural residents in our further analysis. The sample for this part of the comparison temporarily included non-urban populations. Otherwise, our study was restricted to the urban sample.

---

## [Editor Report · Decision Letter 2]

10 Aug 2022

Does political participation help improve the life satisfaction of urban residents: Empirical evidence from China

PONE-D-22-02512R2

Dear Dr. Wang,

We’re pleased to inform you that your manuscript has been judged scientifically suitable for publication and will be formally accepted for publication once it meets all outstanding technical requirements.

Kind regards,

Rong Zhu, Ph.D.

Academic Editor

PLOS ONE

---

## [Editor Report · Acceptance letter]

26 Sep 2022

PONE-D-22-02512R2 

Does political participation help improve the life satisfaction of urban residents: Empirical evidence from China 

Dear Dr. Zhu:

I'm pleased to inform you that your manuscript has been deemed suitable for publication in PLOS ONE. Congratulations! Your manuscript is now with our production department. 

Kind regards, 

on behalf of

Dr. Rong Zhu 

Academic Editor

PLOS ONE